# CRISPR/Cas9 Technology Providing the Therapeutic Landscape of Metastatic Prostate Cancer

**DOI:** 10.3390/ph17121589

**Published:** 2024-11-26

**Authors:** Jieun Park, Jaehong Kim

**Affiliations:** 1Department of Neurology, College of Medicine, Dongguk University, Ilsan, Goyang 10326, Republic of Korea; jieun_park@dongguk.edu; 2Department of Biochemistry, College of Medicine, Gachon University, Incheon 21999, Republic of Korea

**Keywords:** metastatic prostate cancer (mPCa), gene therapy, synthetic lethality, CRISPR/Cas9, DNA damage repair (DDR)

## Abstract

Prostate cancer (PCa) is the most prevalent malignancy and the second leading cause of cancer-related death in men. Although current therapies can effectively manage the primary tumor, most patients with late-stage disease manifest with metastasis in different organs. From surgery to treatment intensification (TI), several combinations of therapies are administered to improve the prognosis of patients with metastatic PCa. Due to the high frequency of the mutation during the metastatic phase, the clustered regularly interspaced short palindromic repeats (CRISPR)/CRISPR-associated nuclease 9 (Cas9) genetic engineering tool can accelerate the effects of TI by enhancing targeted gene therapy or immunotherapy. This review describes the genetic background of metastatic PCa and how CRISPR/Cas9 technology can contribute to the field of PCa treatment development. It also discusses the current limitations of conventional PCa therapy and the potential of CRISPR-based PCa therapy.

## 1. Introduction

Prostate cancer (PCa) is the second most diagnosed solid-organ cancer in men due to its high prevalence [1,2]. The GLOBOCAN 2020 report showed there were 1,414,259 new cases of PCa and 375,304 deaths [3]. Although the incidence of PCa has remained stable from 2014 to 2018, its prevalence accounts for 29% of all malignancies, and since 2011, the incidence of advanced PCa in the USA has been rising by 4–6% each year. Additionally, the prevalence of PCa in men aged >65 years is approximately 60%. Furthermore, the mortality rate of PCa from 2017 to 2021 and the expected number of deaths in 2024 in the USA are 18.8% and 35,250, respectively [4].

More than 95% of PCa cases are adenocarcinomas, with an acinar origin being more common than a ductal origin. Additionally, almost 80% of PCa cases develop from the luminal or basal (with a lesser prevalence) epithelial cells in the peripheral regions that occupy >70% of the prostate gland.

Around 80% of patients are localized PCa patients [5]. If PCa is diagnosed at an early stage, life expectancy may be as high as 99% for >10 years [6]. Data from the Cancer of the Prostate Strategic Urological Research Endeavor registry showed that despite conducting PSA screening, approximately 40% of new cases manifest with intermediate-risk localized disease [7].

Furthermore, 8% of men with PCa have distant metastases (often in multiple sites), while 13% present with locoregional metastases. If PCa is diagnosed when distant metastases have occurred, the overall survival rate is only 34% for 5 years [8]. Metastatic PCa (mPCa) accounts for >400,000 deaths annually and is expected to increase by two-fold or more by 2040 [9].

Although PCa is usually diagnosed at an early stage, the risk–benefit ratio of treatment remains uncertain. The treatment of PCa is one of the most challenging due to the significant morbidity that results from therapy [10,11]. As approximately 20–30% of patients develop metastases, and given that development of metastatic castration-resistant prostate cancer (mCRPC) results in drug resistance, it is important to study the mechanisms of PCa metastasis to overcome drug resistance as well as to personalize therapy. Due to the high mutation burden of mPCa, identifying and targeting genes that induce metastasis are important for advancing personalized medicine. CRISPR/Cas9 technology offers a platform to detect metastasis drivers and provides tools for clinical treatment through gene therapy. This review focuses on the comprehensive analysis of the cause of mPCa and the latest developments in its treatment, including experimental trials in PCa research. Additionally, this review also includes a brief introduction to CRISPR technology and how it can be employed in PCa research. All the data were prepared by searching the literature in PubMed, clinicaltrials.gov, and Web of Science.

## 2. Biology of Metastatic PCa

mPCa is a serious health issue due to the increasing prevalence of advanced disease as well as its effects on quality of life and as a cause of mortality. PCa metastasis is mostly associated with spread to the locoregional lymph nodes and/or hematogenously to the stroma of the bone marrow [10]. Uncommonly, PCa metastasis is associated with spread to distant visceral sites. Overall, >80% of distant metastatic lesions occur in bone tissues [10], with osteoblastic bone lesions in the axial skeleton being the most common metastatic sites in advanced PCa [12]. Most bone metastasis shows mixed characteristics of osteoblastic lesions with osteolytic components, and patients experience severe pain, hypercalcemia, and frequent fractures (Figure 1).

Tumor cells undergo epithelial–mesenchymal transition (EMT), leave the primary site, and enter the circulation as circulating tumor cells (CTCs). Only a small proportion extravasates at a distant site and persists as disseminated tumor cells (DTCs). Prostate cancer cells metastasize to bone through a complex interplay of factors, including the expression of specific adhesion molecules and chemokine receptors, such as CXCR4, which direct cancer cells to the bone microenvironment [13]. When cancer cells invade the bone marrow, the interaction between the cancer cells and the bone microenvironment leads to a “vicious cycle” of bone formation and resorption, which eventually leads to cancer cell survival and tumor growth [14]. This process is critical to the progression to metastatic castration-resistant prostate cancer (mCRPC), as it enables PCa cells to thrive despite androgen deprivation. PCa cells also compete with hematopoietic stem cells (HSCs) for the occupancy of limited niches in the bone marrow [15,16], and the reduction of the niche size hampers dissemination [17]. Once DTCs occupy the vascular niche, they acquire a stem-cell-like phenotype [18]; together with the protective microenvironment, this results in DTCs that are highly resistant to therapy [19,20].

Almost all patients with mPCa experience castration-resistant PCa (CRPC) that is refractory to androgen deprivation therapy (ADT), which is the primary cause of morbidity and mortality [10]. mCRPC eventually becomes therapy- and castration-resistant PCa (t-CRPC), which is considered end-stage disease due to the unavailability of effective treatment options [11,12].

## 3. Genetics of Metastatic PCa

Many genetic factors are involved in PCa metastasis. Early-stage PCa has a relatively lower frequency of point mutations compared to other cancers and includes large-scale chromosomal rearrangements such as ETS family gene fusion [21]. The most common genetic alteration in mPCa is gene fusion between the androgen receptor (*AR*)-regulated transmembrane serine protease (*TMPRSS2*) and transcription factor erythroblast transformation-specific (*ERG*) genes (41–43% in the case of mPCa, Table 1). *TMPRSS2–ERG* fusion belongs to the ETS family rearrangement and accounts for 90% of the total ETS family fusions [22]. Although *TMPRS–ERG* fusion is strongly correlated with the stage and prognosis of PCa [23,24], the significance of *TMPRSS2–ERG* fusion in the tumorigenesis of PCa remains unknown [25,26]. This gene fusion upregulates ERG expression and reactivates AR signaling in tumor cells, with amplifications and/or mutations of AR strongly correlating with the onset of mPCa [5].

The *AR* gene in chromosome X (Xq11-12) is the most researched molecular factor in PCa research and reportedly promotes CRPC. AR is a ligand-dependent nuclear transcription factor; binding to its ligands, namely testosterone or dihydrotestosterone (DHT), results in the transcription of AR-responsive genes that induce proliferation and promote survival of prostate epithelial cells. Approximately 20% of patients with CRPC have X chromosome rearrangement and subsequent AR amplification, resulting in increased levels of AR proteins in tumor cells [27]. Alterations in AR signaling are the drivers of resistance to ADT in patients with mCRPC [9].

Recurrent hotspot mutations in the speckle-type BTB/POZ protein (*SPOP*) (~10%), forkhead box A1 (*FOXA1*) (~5%), phosphatase and tensin homolog (*PTEN*) (40%), and tumor protein 53 (*TP53*) (~50%) are also enriched in patients with mCRPC (Table 1) [28,29].

**Table 1 pharmaceuticals-17-01589-t001:** Frequency of somatic and germline mutations by prostate cancer stage. Reprinted from Sandhu et al. [9], Copyright 2021, with permission from Elsevier. * Castration sensitivity was not defined in this study.

Somatic Mutations	Localized (*n* = 333) [30]	Metastatic Castration Sensitive (*n* = 140) [31]	Metastatic Castration Resistant (n = 444) [32] and (*n* = 101) [33]
*TMPRSS2-ERG* fusion	46.0%	Not reported	41.0% and 43.0%
Other *ETS* family gene fusions	14.0%	Not reported	10.0% and 15.0%
*SPOP* mutation	11.0%	11.0%	5.0% and 6.0%
*CHD1* deletion	7.0%	6.0%	23.0% and 33.0%
*FOXA1* mutation	4.0%	10.0%	9.0% and 19.0%
*PTEN* deletion	17.0%	17.0%	32.0% and 45.0%
*TP53* mutation or deletion	8.0%	30.0%	40.0% and 57.0%
*RB1* deletion	1.0%	7.0%	12.0% and 13.0%
*PI3K* mutation	3.0%	5.0%	5.0% and 5.0%
*AKT* mutation	1.0%	2.0%	1.0% and 2.0%
*BRCA1* mutation or deletion	1.0%	1.0%	1.0% and 2.0%
*BRCA2* mutation or deletion	3.0%	7.0%	10.0% and 11.0%
*ATM* mutation	1.0%	2.0%	1.0% and 2.0%
*CDK12* mutation	2.0%	6.0%	3.0% and 7.0%
Mismatch repair mutation	5.0%	5.0%	4.0% and 5.0%
*APC* deletion	5.0%	13.0%	8.0% and 9.0%
*CTNNB1* mutation	2.0%	6.0%	4.0% and 6.0%
*MYC* gain of function	7.0%	6.0%	23.0% and 33.0%
*AR* amplification or mutation	1.0%	4.0%	59.0% and 70.0%
**Germline Mutations**	**Localized (*n* = 499)** [34]		**Metastatic * (*n* = 692)** [34]
*BRCA1*	0.6%		0.9%
*BRCA2*	0.2%		5.3%
*ATM*	1.0%		1.6%
*CHEK2*	0.4%		1.9%
*PALB2*	0.4%		0.4%
*RAD51D*	0.4%		0.4%
Mismatch repair (Lynch syndrome)	0.6%		0.6%

*SPOP* encodes an E3 ubiquitin ligase, and its mutation prevents the degradation of the ERG and AR proteins [35]. SPOP also acts as a negative regulator of PCa cell proliferation through the activation of both phosphatidylinositol 3-kinase (PI3K)-AKT serine/threonine kinase (AKT)-mammalian target of rapamycin (mTOR) and AR signaling [36], with SPOP-mutated PCa cells being resistant to bromodomain and extra-terminal motif (BET) inhibitors [37]. Studies showed that the *SPOP* mutation sensitizes cancer cells not only to AR inhibitors but also to poly (ADP-ribose) polymerase inhibitors (PARPi) by repressing homology recombination (HR) and promoting non-homologous end-joining (NHEJ) DNA repair [38].

*PTEN* mutation is another hallmark of human malignancies and is a key determinant of metastasis. PTEN suppresses the PI3K-AKT-mTOR signaling pathway, which regulates cell proliferation and energy metabolism [39].

Loss-of-function mutations in the cyclin-dependent kinase 12 (*CDK12*) gene represent a specific subtype of mCRPC [40]. Compared with primary PCa, mCRPC is enriched with *CDK12* mutations (6.9% vs. 1.2% of 360 vs. 498 patients) that mostly harbor a truncated kinase domain (amino acids 728–1020) [40,41]. CDK12 is involved in (i) regulation of RNA polymerase II transcription by phosphorylating serin residues of the hepta-peptide repeats (YSPTSPS) in the C-terminal domain of RNAPII, which allows entry into the elongation phase of transcription [42], and (ii) regulation of expression of DNA damage repair (DDR) genes (*BRCA1, FANCD2, FANCJ, ATR*) [43]. CDK12 loss is mutually exclusive from other primary genetic drivers (PGDs) such as ETS fusion, SPOP mutations, and mismatch-repair (MMR) deficiency, and it is associated with a high genome-wide frequency of focal tandem duplications [40]. A study also described the distinct pattern of *CDK12*-mutated mCRPC, showing the high chromosomal breakage numbers by exome sequencing and the worse prognosis compared to controls [44].

Aside from PGDs, mutations in DNA damage repair (DDR) genes like *BRCA2* and *ATM* increase genomic instability, making prostate cancer more aggressive and often leading to poorer responses to standard therapies. DDR pathway-related genes are highly mutated in 655 patients with mCRPC, as revealed by multi-institutional clinical sequencing projects [41]. A report from the International Stand Up to Cancer/American Association for Cancer Research Prostate Cancer/Prostate Cancer Foundation Team (SU2C-PCF) showed genetic alterations of DDR genes in 23% of 150 metastatic biopsy samples [45]. In 2018, the prevalence of MMR defects in PCa was established in a large series involving 1033 patients. Inherited mutations in genes involved in MMR, namely *MLH1, MSH2*, and *PMS2*, also increase the risk of PCa [46]. Another recent publication showed the impact of germline *BRCA1/2* and *CHEK2* mutations in 76 mPCa patients receiving ADT [47]. The results showed that the median time to CRPC was significantly shorter in carriers of *BRCA2* and *CHEK2* mutations compared to non-carriers, indicating that those mutations cause faster progression to CRPC along with the hormonal therapy.

HR pathway alterations are early events during the evolution of aggressive PCa. The Cancer Genome Atlas (TCGA) reported the molecular analysis of 333 primary prostate tumors, with 19% of them harboring alterations in DDR genes, including *BRCA2, BRCA1, ATM, CDK12*, *FANCD2*, or *Rad51C* [30]. *BRCA2*, which is a critical regulator of the HR repair pathway, is the most frequently mutated DDR gene in PCa. In total, 13.3% of patients with advanced PCa harbor *BRCA2* alterations, with the *BRCA2* mutations resulting in sensitivity to PARPi treatments [48,49]. A report on 1211 men with PCa who were undergoing active surveillance, including 11, 11, and 5 with *BRCA1, BRCA2*, and *ATM* germline carriers, respectively, revealed that *BRCA2* carriers are more likely to undergo a tumor grade re-classification in subsequent biopsies [50]. Another retrospective study that evaluated the outcomes of 1302 patients reported that after radical treatment, *BRCA1/2* carriers developed metastasis significantly earlier than non-carriers (13.2 vs. 28 months, *p* = 0.05) [51]. A prospective study also reported that among 53 patients with de novo metastatic hormone sensitive prostate cancer (mHSPC), the time to castration resistance (TTCR) was significantly shorter in 11 cases with somatic and/or germline DDR alterations. Men with germline *BRCA1* or *BRCA2* mutations have a three- to eight-fold higher lifetime risk of PCa that can behave aggressively because of additional MYC activation in combination with inactivation of TP53 and PTEN [52,53].

## 4. Current Standard Treatments

An important characteristic of PCa is its hormone responsiveness. Similar to normal prostate cells, PCa cells need androgens for growth [54]. Hence, the primary treatment for advanced or metastatic PCa is ADT through surgical or pharmacological castration [55]. A decrease in testosterone levels is achieved by surgical removal of the testicles or treatment with luteinizing hormone-releasing hormone agonists, anti-androgens, and estrogens. Although ADT reduces the severity of symptoms and attenuates tumor growth, ADT resistance can develop, leading to mCRPC recurrence. Therefore, single-drug treatments should not be considered for mPCa [31].

Enzalutamide, which is a potent second-generation AR antagonist, is used for mCRPC therapy, resulting in a significant improvement in patient survival rates [56]. A large-scale randomized trial reported that enzalutamide extends the time to metastasis and increases the overall survival rates in patients with non-mCRPC [57]. However, most patients eventually developed resistance to enzalutamide, warranting alternative therapies. AR-independent enzalutamide-resistant mechanisms are characterized by the bypassing of AR signaling via other hormone nuclear receptors, such as the glucocorticoid receptor [58], or by developing lineage plasticity traits through the expression of neuroendocrine and stem-cell-related genes [59,60].

Multiple therapeutic strategies in combination with AR antagonists were tested. Enzalutamide plus abiraterone acetate was tested in patients with resistance to enzalutamide (NCT01995513) [61]. Abiraterone acetate with apalutamide and prednisone was tested successfully in mCRPC patients (NCT02257736) [62]. ADT with apalutamide was successfully tested in metastatic castration-sensitive PCa patients (NCT02489318) [63]. Abiraterone acetate combined with ADT plus prednisone significantly extended progression-free survival (NCT01715285) [64]. EZH2 inhibitors are under investigation with abiraterone and enzalutamide (NCT03480646) or with the AR antagonist (NCT03741712) in the treatment of mCRPC. For combination therapeutic studies with radiotherapy, a previous review article summarized the latest available clinical trials for patients with localized or locally advanced prostate cancer, which could potentially be applied to mPCa patients in terms of TI [65].

Palliative treatment is essential for patients with bone metastasis and should aim to relieve pain, enhance mobility, and prevent complications such as pathologic fractures or epidural cord compression [66]. Surgical resections of the spinal cord show the heterogeneity of bone metastasis [67,68]. It should be noted that nuclear AR-negative tumor cells are present in both CRPC and treatment-naïve mPCa [69]. The heterogeneity of mPCa regarding AR signaling indicates second-generation AR-directed therapies such as abiraterone and enzalutamide [70], and it will most likely require additional therapies, such as bone-targeting therapies and those directed against non-AR pathways.

## 5. Inhibition of DNA Repair as a Targeted Gene Therapy for mPCa

Due to the significant mutational burden of mPCa, targeted gene therapy offers a complementary approach to ADT for patients with mCRPC, wherein the concept of synthetic lethality can be employed (Figure 2).

Cancer cells harbor mutated genes; if partner genes are suppressed by inhibitors, specific cancer cell death may occur while sparing normal cells [71]. The first synthetic lethality-targeting drugs were PARP inhibitors for BRCAness patients [72]. PARP1 senses DNA lesions, such as single-strand DNA break (SSB) and double-strand DNA break (DSB), inducing self-activation through poly(ADP-ribosy)lation (PARylation). PARylated PARP1 then recruits other DDR factors and promotes downstream signaling of repair pathways [73].

Various genomic studies have reported that 15–35% of mCRPC cases harbor DNA repair defects, including in BRCA1/2, ATM, ATR, and RAD51 (TCGA Research Network 2015) [41]. Germline mutations in *BRCA* genes are correlated with an increased risk of PCa development or a more aggressive phenotype as well as worse outcomes [74,75]. Currently, olaparib, rucaparib, niraparib, and talazoparib are the only FDA-approved PARPi in the USA. These inhibitors trap PARP1 and PARP2 at SSBs that result in stalled and collapsed replication forks. Consequently, SSBs are converted into DSBs, resulting in inefficient repair by HR-deficient cells and causing catastrophic DNA damage, cell cycle arrest, and cell death of tumors [76]. Olaparib and rucaparib are approved by the FDA for mCRPC with deleterious germline and/or somatic mutations in *BRCA1/2* [48,49]. In phase II and III trials, olaparib for mCRPC resulted in high response rates, as evidenced by prolonged progression-free or increased overall survival rates [77,78].

Beyond PARPi, extensive research has been conducted to develop synthetic lethality that targets other metastasis drivers in cancer cells. PTEN loss is another hallmark of mPCa that hyperactivates PI3K/AKT signaling and stimulates tumor cell survival and metastasis in vitro. In a genetically engineered murine model, PTEN loss cooperated with RAS/MAPK signaling to promote EMT and macro metastasis [79]. Various reports suggest the synthetic relationship between *PTEN* and other genes. Zhao et al. conducted a large-scale genomic analysis of the TCGA database and reported that CHD1 is in a synthetic lethality relationship with PTEN deficiency. Functional PTEN promotes degradation of CHD1, whereas PTEN-deficient PCa shows stabilization of CHD1 and activation of the pro-tumorigenic TNF-NFκB signaling pathway [80]. These findings indicate trackable synthetic lethal targets in PTEN-deficient PCa. CDK12 loss is another subset of the mutational profiles of mPCa patients [40]. Wu et al. conducted an integrative genomic analysis of data from 360 patients with mCRPC and revealed that CDK12 loss defines another subtype of mPCa that enables the application of the checkpoint inhibitor anti-PD1 as a treatment in these patients [40].

Large-scale genomic analyses have been performed to reveal PGDs for targeted therapy. As genome-wide CRISPR/Cas9 screening can be employed, the identification of targetable gene alterations required for cancer cell survival and the development of a synergistic treatment with existing therapies have become feasible [81].

## 6. CRISPR Technology for mPCa Therapeutics

A remarkable number of patients with mCRPC harbor defects in genes involved in the DDR pathway. Additionally, a significant proportion of alterations are present in the germline. Through genome-editing tools, gene therapy has been developed and improved over the past few decades. CRISPR/Cas9 technology is one of the tools utilized in precision medicine that has the potential to be employed in cancer therapy due to its high accuracy and efficiency in terms of gene alteration.

CRISPR/Cas9 is comprised of the Cas9 enzyme and guide RNA (gRNA). The Doudna group first synthesized single gRNA (sgRNA) that can target a specific DNA sequence and purified the Cas9 enzyme that cleaves DNA at a desired location [82]. Target binding is driven by an sgRNA, and the gRNA/Cas9 RNP complex hybridizes to an intended DNA region containing the sequence (protospacer) complementary to the gRNA and protospacer-adjacent motif (PAM). Through this method, it is possible to perform DNA editing, including insertion, deletion and modification at the level of single base pairs [83,84]. This programmable gene-editing technology revolutionized various fields, including medicine and agriculture (Figure 3A).

A representative Cas9-engineering model is catalytically deactivated Cas9 (dCas9), which is mutated on two amino acids (D10A/H840A) of *Streptococcus pyogenes* Cas9 (SpCas9) [83]. dCas9 acts as a locator that searches for specific genomic loci under sgRNA guidance, and it can be conjugated to other effector proteins that perform enzymatic functions on the genome differently. For example, the CRISPRa (activation) and CRISPRi (interference) systems are composed of dCas9 fused to a transcription activator and repressor, respectively [83] (Figure 3B). Along with the CRISPR KO library, these screening tools can be employed to not only multiply activate or repress target genes under certain conditions, such as during therapy, but also identify genes required for cancer cell survivals as candidate targets.

To improve the application of CRISPR technology, various delivery systems have been developed. Adeno-associated virus (AAV) vector- and polyethylenimine (PEI)-derived graphene quantum dot (PEI-GQD)-based CRISPR delivery systems hold promise for targeted therapy in metastatic prostate cancer. AAV vector provides high specificity but is limited by the delivery efficiency and immune response [85]. PEI-GQD is a newer approach that can facilitate intracellular delivery but requires further development to ensure its safety and targeting accuracy in clinical applications [86].

The recent applications of the CRISPR system can be categorized into three groups: mechanism of drug resistance, metastasis, and treatment (Table 2). Various CRISPR-based strategies have been proposed, including (i) suppression of oncogenes or repair of genetic mutations such as *BRCA1* and *BRCA2* mutations [87], and (ii) enhancement of the immune response to cancer cells by engineering T cells using CRISPR technology.

### 6.1. Drug Resistance Mechanism

Although AR signaling and PARP inhibitors prolong the progression-free and overall survival of patients with mPCa, drug resistance frequently develops and has become a serious concern. To overcome drug resistance, CRISPR/Cas9 technology can be employed to identify novel targets that can synergize the treatment effects using conventional mono-treatment.

Some germline alterations, specifically those involved in HR, can act as predictors of the response to PARP inhibition. Tsujino et al. revealed that the loss of checkpoint kinase 2 (CHEK2) confers resistance to PARPi through the upregulation of BRCA2 expression [88]. They also conducted CRISPR KO screening in four BRCA1/2-deficient PCa cell lines (LNCaP, C4-2B, 22Rv1, and DU145) with and without olaparib. The CRISPR KO library targets over 18,000 protein-coding genes, and negatively and positively selected gene KOs confer sensitivity and resistance, respectively, to olaparib. Gene Ontology (GO) analysis revealed that 67 negatively selected common heats shared by at least two cell lines are related to DNA repair and replication. Meanwhile, 103 positively selected genes that are shared by at least two cell lines are enriched in cell cycle phase transition and positive regulation of gene expression. Among them, a loss of MMS22L, which is a component of TONSL that recognizes and repairs DSB at stalled or collapsed replication forks [95], increases the PARPi response due to impaired HR function, and its effect is dependent on TP53.

Furthermore, they also revealed that the loss of CHEK2 enhances HR function through E2F7-controlled BRCA2 expression, resulting in olaparib resistance. *CHEK2* is a BRCAness gene due to its phosphorylation of the *BRCA1* gene that promotes HR, and it has been utilized as a biomarker for olaparib treatment in clinical trials [49,96]. Hence, olaparib resistance conferred by CHEK2 loss in mCRPC cell lines was a surprising result. This finding indicates the value of the proper use of CRISPR/Cas9 screening as it revealed novel information that can contradict to traditional perspectives.

Regarding AR signaling inhibitors, Lei et al. utilized CRSIRP/Cas9 screening under AR suppression with enzalutamide treatment, revealing that CDK12 is required for PCa cell survival while its inhibition suppresses proliferation and induces apoptosis of PCa cells [89]. Although this finding is inconsistent with previous reports regarding CDK12 loss-of-function mutations, the synergistic anti-PCa effect is obvious when a CDK12 inhibitor and AR antagonists are combined. The effects may be due to attenuated H3K27ac signaling on AR targets and intensive super-enhancer-associated apoptosis pathways. Notably, that was the first report that showed CDK12 to be a conservative target of PCa using the CRISPR/Cas9 screening system and that CDK12 may be a potential therapeutic target for PCa treatment.

Liu et al. also utilized CRISPR KO screening to identify casein kinase 1α (CK1α) as a therapeutic target to overcome enzalutamide resistance in mPCa [90]. Depletion or inhibition of CK1α stabilized the ATM protein through phosphorylation and activated downstream DDR signaling, resulting in sensitization to enzalutamide.

### 6.2. Metastasis Drivers

Several studies have employed CRISPR technology to identify key drivers of metastasis in PCa. Cai et al. developed a mouse model that allows the development of simultaneous and multiple gene mutations in the epithelia of the prostate [91]. To observe the effects of the conditional loss of specific genes in the mouse prostate, intercrossing of multiple mouse strains, which is extremely laborious and time consuming, was previously necessary. However, CRISPR with an adeno-associated virus (AAV) delivery system allows for the simultaneous mutation of five tumor suppressors (TP53, PTEN, Rb1, Stk11, and RnaseL), resulting in the creation of a rapid, invasive, and androgen-independent tumor mouse model. Three additional gene knockouts (*Zbtb16, KMT2C, and Kmt2d*) showed that the loss of KMT2C was essential to induce lung metastasis but not tumor progression. KMT2C is a histone methyltransferase found to be mutated in many types of cancers [41,97], but its role in PCa remained elusive. This study not only provides a novel PCa mouse model for CRPC but also suggests new factors required for tumor progression and metastasis.

Another study revealed a molecular driver of bone metastasis in PCa using CRISPR [92]. In that study, genome-wide CRISPRa (activation) and CRISPRi (inhibition) libraries were generated and each library contains 5 sgRNAs targeting 18,915 genes and 1895 non-targeting control sgRNAs in non-metastatic human PCa cell line 22Rv1. The nuclease-dead dCas9 was coupled to a transcriptional activator (sunCas9) or repressor (dCas9-KRAB) and integrated into 22Rv1 labeled with GFP-luciferase [98]. Genome-wide sgRNA libraries were then packaged into lentiviruses that infected the target cell, followed by implantation into mice. The subsequent development of metastatic tumors was visualized by the GFP signals, and tumor samples were sequenced to identify enriched sgRNAs. Using this system, it was revealed that the CITED2 gene is a driver of bone metastasis in PCa. CITED2 is a transcriptional co-activator that promotes metastasis in other cancers [99,100]. The results successfully confirmed the possible role of CITED2 in PCa metastasis using the CRISPR system.

Exploring the role of non-coding RNA in mPCa CRISPR has been attempted. Extensive studies have reported the approach for the regulation of mPCa metastasis using microRNA (miRNA) treatment: miR-34a exhibits anti-prostate cancer stem cells effect by targeting invasiveness and metastasis [101]. Effective delivery methods for miR-34a have also been critically developed [102,103]. Camargo et al. revised metalloproteinase 9 (MMP9) and microRNA (miR) miR-21, revealing that they attenuate PCa metastasis [93]. The ECM-degrading enzyme MMP9 contributes to the infiltration of tumor cells into other organs. Thus, alteration of MMP9 expression may contribute to PCa evolution and affect its metastatic potential [104]. miR-21, which upregulates MMP9, is highly expressed in PCa, and its inhibition reduced metastasis in a PCa xenograft model, leading to downregulation of reversion-inducing cysteine rich protein with Kazal motifs (RECK) signaling [105]. miR-21 also regulates B-cell translocation gene 2 (BTG2), which is linked to PCa progression [106], and myristoylated alanine-rich protein kinase c substrate (MARCKS), which controls cellular invasion [107]. sgRNAs targeting MMP9 and miR-21 sequences were inserted into a PX-330 plasmid and transfected into DU145 and PC-3, resulting in the attenuation of cell proliferation and invasion and inducing apoptosis through the upregulation of RECK expression.

### 6.3. Treatment

For targeted therapy, CRISPR/Cas9 screening can be used in mPCa to identify novel targets to induce synthetic lethality. Ding et al. performed CRIPSR KO screening with or without PTEN knockdown to determine epigenetic regulators that induce synthetic lethality with PTEN deficiency in mPCa [94]. Their results revealed that SWI/SNF subunit Brahma-related gene 1 (*BRG1*) (SMARCA4) knockdown results in decreased cell proliferation of PTEN-negative cells (LNCaP, C4-2, and PC3) but not PTEN-competent cells (22Rv1, BPH-1, and LAPC4). In a PTEN-null pre-clinical model, treatment with a BRG1 antagonist inhibited the progression of PTEN-deficient PCa. Mechanistically, upregulated BRG1 expression in a PTEN-deficient cell line causes chromatin remodeling, thereby stimulating pro-tumorigenic transcription.

Delivery of the CRIPSR system into the human body is another concern in terms of the treatment efficiency. Although an AAV viral delivery system may be the most promising candidate due to its reduced risk of genomic integration, inherent tissue tropism, and clinically manageable immunogenicity, it may cause carcinogenesis and has a limited loading capacity and restricted scalability for use in the human body. Therefore, the non-viral delivery system has been extensively developed. Lee et al. developed a nanomaterial PEI-GQD-CRISPR RNP system to overcome physiological barriers and enable the visual tracking of genes of interest [86]. GQD is highly scalable due to the ease of synthesis and widely available precursor materials. With this delivery system, TP53 gene mutations in the PC-3 cell line were successfully converted into wildtype, cancer cell viability was dramatically reduced to 60%, and the increase in the apoptosis signal was similar to that following staurosporine treatment (apoptosis inducer) without significantly affecting HEK293T cells. These results suggest a promising avenue for GQD delivery of CRISPR-Cas9 therapeutics in vivo.

## 7. Conclusions and Future Directions

Although there are no on-going clinical trials using CRISPR technology in PCa, CRISPR may potentially be used to treat PCa based on its successful application in other cancers, with immunotherapy being the most recent development using CRISPR technology.

The remarkable effects of chimeric-antigen receptor (CAR)-T cell therapies in hematological malignancies have inspired its development and use for the treatment of mCRPC. As solid tumors have an immunosuppressive tumor microenvironment (TME) [108,109], strategies to improve the function of T cells should be developed for successful immunotherapy. Clinical trials of CAR-T cells against solid tumors, including PCa, are being performed, and several studies have already shown the possibility of CAR-T cell therapy in mCRPC by engineering T cells [110,111].

CRISPR technology can improve the persistence of CAR-T cells to target and kill cancer cells [112]. A breakthrough discovery in cancer research is the blockade of interaction between the programmed cell death protein (PD-1) on T cells and the programmed cell death ligand (PD-L1) on tumor cells [113,114]. In a syngeneic immunocompetent mouse model, Dötsch et al. revealed that CRISPR/Cas9-mediated PD-1 KO CD19-CAR-T cells are continuously exposed to antigens and survive for over 390 days [115]. Another study revealed that CRISPR-mediated LAG-3 knockout CAR-T cells displayed robust antigen-specific antitumor activity in a cell culture in vitro and in a mouse xenograft model [116].

Although there are some challenges in using CRISPR technology in humans, such as off-target effects and cytotoxicity from unwanted DNS cleavages, the development of base editing or prime editing, which do not involve DSBs, shows potential to overcome these ethical concerns [117].

CRISPR/Cas9 technology can potentially enhance targeted gene therapy as well as immunotherapy. It can also reveal novel genetic alterations in mPCa that can be targeted to improve the effectiveness of CAR-T therapy through genetic engineering. Due to its high accuracy and efficiency, CRISPR/Cas9 technology has advantages over other genetic engineering tools for personalized medicine. CRIPSR/Cas9 is expected to offer new hope to PCa patients by providing them with effective and affordable personalized treatment.

## Figures and Tables

**Figure 1 pharmaceuticals-17-01589-f001:**
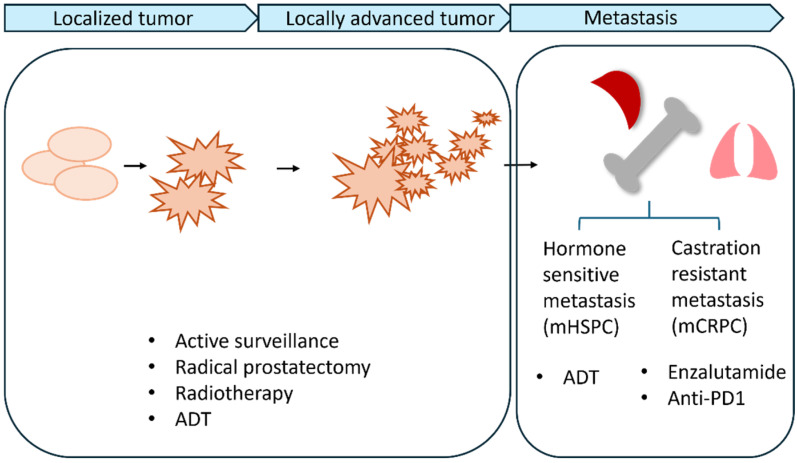
Progression of prostate cancer and the development of mCRPC. Localized adenocarcinoma can progress to invasive carcinoma and spread to distant organs such as the lymph nodes, bones and lungs. Standardized treatments are effective in the early stage of cancer, but many metastatic patients develop drug resistance and experience a significant increase in the mutational burden. Therefore, a combination of standard treatment with gene therapy could enhance the overall prognosis of mPCa patients.

**Figure 2 pharmaceuticals-17-01589-f002:**
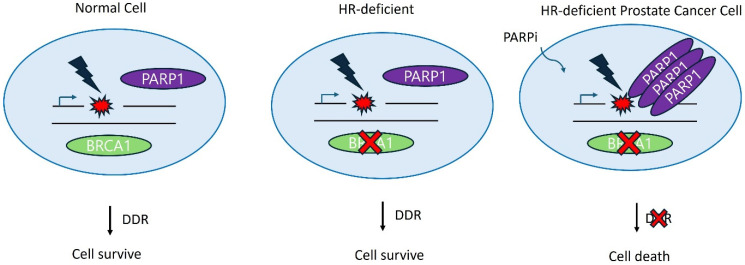
Schematic diagram of the synthetic lethality in cancer cell treatment and the mode of action of PARP inhibitors. In HR-deficient cells, a single alteration of the DDR gene (*BRCA1*) can allow survival, whereas simultaneous alterations in both partner genes (*BRCA1* and *PARP1*) by application of PARP inhibitors can lead to the death of HR-deficient prostate cancer cells. PARP inhibitors work by trapping PARP proteins at the site of DNA damage, which prevents proper DDR and ultimately results in cancer cell death.

**Figure 3 pharmaceuticals-17-01589-f003:**
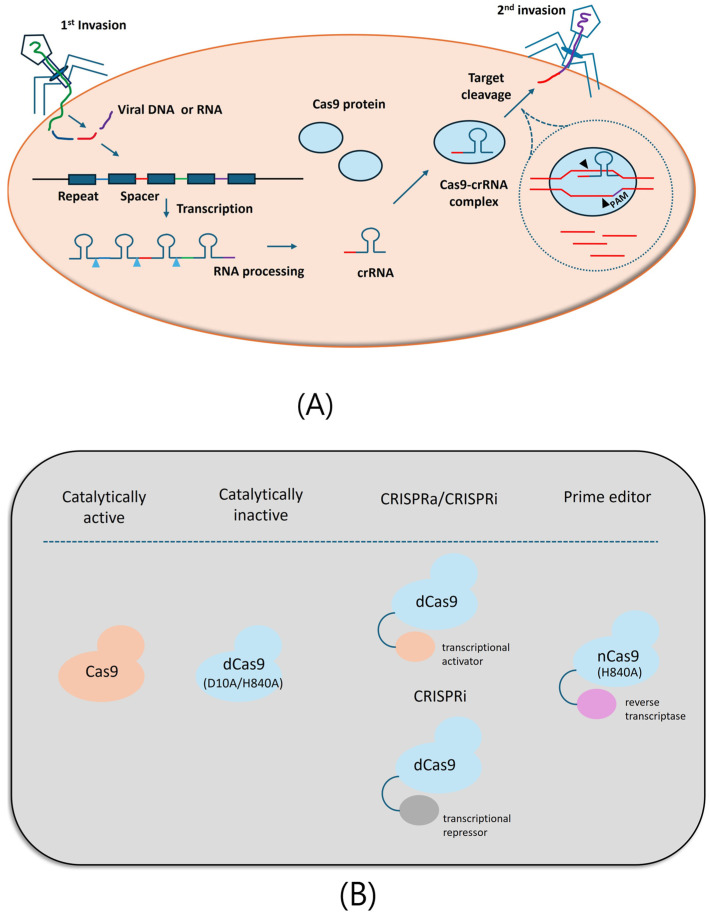
CRISPR-Cas9 system. (**A**) Mechanism of the bacterial immune defense system CRISPR/Cas9. Bacteria develop an adaptive immune system to defend themselves from virus invasions. (i) Adaptation: the first vial genetic materials (spacer: blue, red, green, purple lines) are integrated into the host genome within the CRIPSR array, separated by palindromic repeats (repeat: dark blue boxes). (ii) Expression: the CRISPR array is transcribed into RNA, followed by RNA processing, generating CRISPR RNAs (crRNA). (iii) Interference: when the second vial invasion occurs, crRNA guides a bacterial CRISPR-associated protein 9 (Cas9) protein to the viral DNA/RNA and cleaves it to deactivate. This mechanism is applied to the generation of the revolutionary gene-editing tool, CRISPR/Cas9 technology, to modify the DNA/RNA in various organisms. (**B**) Representative engineering of CRISPR/Cas9 technology. Catalytically active SpCas9 is the most widely used Cas9 as an editing tool. It is mutated to make catalytically dead Cas9 (D10A/H840A) which lacks endonuclease activity but can still bind to DNA. dCas9 acts as a locator for specific genomic loci and it can be fused to different effector proteins such as transcriptional activator (CRISPRa) or repressor (CRISPRi) to multiply regulate the target gene expressions. The most recently developed editing tool is a prime editor that is composed of nickase Cas9 (H840A) fused to reverse transcriptase and edits the DNA at the single base level without double strand breaks (DSBs).

**Table 2 pharmaceuticals-17-01589-t002:** The recent applications of CRISPR technology in the field of prostate cancer research.

Subject	Organism	Target	Methods	Genetic Factors		Ref.
**Drug** **resistance mechanism**	In vitro, in vivo	PARP inhibitor sensitivity and resistance	CRISPR KO library	*MMS22L* KO	*CHEK2* KO	Tsujino et al. [88] (2023)
				Increase of sensitivity to PARPi	Increase of resistance to PARPi	
	In vitro	AR inhibitor resistance	CRISPR KO library	*CDK12* KO		Lei et al. [89] (2021)
				Synergistic effect with ARi		
	In vitro, in vivo	AR inhibitor resistance	CRISPR KO library	*CK1α* KO		Liu et al. [90] (2023)
				Increase of sensitivity to ARi		
**Metastasis drivers**	In vivo	Lung metastasis	CRISPR KO library	*KMT2C*		Cai et al. [91] (2024)
				Driver of lung metastasis		
	In vivo	Bone metastasis	CRISPRa/ CRISPRi library	*CTIED2*		Arriaga et al. [92] (2024)
				Driver of bone metastasis		
	In vitro	Cancer cell proliferation and migration	CRISPR KO library	*MMP9*, *miR-21*		Camargo et al. [93] (2023)
				Driver of metastasis		
**Treatment options**	In vivo	Synthetic lethal target dentification	CRISPR KO library	*BRG1* KO		Ding et al. [94] (2019)
				Inhibition of PTEN-deficient PCa progression		
	In vitro	Nanotherapeutics Correction of oncogene TP53	PEI-GQD/ CRISPR RNP	*TP53* KI		Lee et al. [86] (2023)
				Induction of apoptotic cell death of prostate cancer cell		

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
