# Peer review of "CRISPR/Cas9 Technology Providing the Therapeutic Landscape of Metastatic Prostate Cancer"

_pharmaceuticals, 2024, doi:10.3390/ph17121589_

Round 1

Reviewer 1 Report

Comments and Suggestions for Authors

The manuscript titled “CRISPR/Cas9 technology providing a therapeutic landscape of metastatic prostate cancer” presents comprehensive overview of metastatic prostate cancer focusing on its genetic background, current treatments and the potential of CRISPR/Cas9 technology in developing new therapies. The authors incorporate recent statistics and findings from reputable sources, providing readers with current information on mPCa prevalence, mortality rates, and genetic alterations. The article offers an in-depth examination of genetic factors involved in mPCa, including common mutations and their implications for disease progression and treatment. The discussion on the application of CRISPR/Cas9 technology in mPCa treatment is particularly noteworthy. The article effectively outlines how this genetic engineering tool can enhance targeted therapies and immunotherapy, which is a cutting-edge area of research in oncology. The potential of CRISPR to address drug resistance and identify metastasis drivers is well articulated, providing a forward-looking perspective on treatment strategies. The authors summarize existing literature and highlight ongoing research and potential future applications of CRISPR technology in mPCa. This includes the exploration of synthetic lethality and the identification of novel therapeutic targets, which are critical for advancing personalized medicine. The article also discusses the clinical implications of CRISPR technology in the context of current treatment paradigms, including the limitations of androgen deprivation therapy and the emergence of castration-resistant prostate cancer. Overall, this article is reasonably well written and provides a valuable overview of metastatic prostate cancer and the potential of CRISPR/Cas9 technology in developing new therapies. The reviewer has the following comments to the authors that need to be addressed

1.     Although this article discusses the potential of CRISPR technology, the inclusion of a comprehensive overview of ongoing clinical trails for new mPCa treatments discussing their effectiveness and limitations would enhance the impact of the manuscript.

2.     The introduction part could benefit from a more focused discussion on the current challenges in prostate cancer therapeutics and the specific gaps this study aims to address. The reviewer recommends including the following latest publications highlighting the advancements in prostate cancer treatment.

https://www.cell.com/molecular-therapy-family/nucleic-acids/fulltext/S2162-2531(24)00080-5

https://www.mdpi.com/1422-0067/25/4/2123

3.     A discussion on the ethical implications of using CRISPR technology in human subjects, particularly concerning potential off-target effects and long-term consequences of genetic modifications would provide a more balanced view of the technology's application.

4.     The article appears to end abruptly in the middle of a section about DNA repair inhibition, suggesting that some content may be missing or incomplete.

 5.      There are some typographical errors and instances of awkward phrasing in the manuscript. The authors should carefully proofread to correct these issues and ensure that all abbreviations are properly defined upon first use. For example, the title of Section 5.1 could also be revised for clarity. 

Author Response

  1. Although this article discusses the potential of CRISPR technology, the inclusion of a comprehensive overview of ongoing clinical trials for new mPCa treatments discussing their effectiveness and limitations would enhance the impact of the manuscript.

Our response: Thank you for your insightful comment. In response, we have added a new paragraph addressing recent clinical trials for mPCa treatments in the section of ‘Current standard treatments’.

  1. The introduction part could benefit from a more focused discussion on the current challenges in prostate cancer therapeutics and the specific gaps this study aims to address. The reviewer recommends including the following latest publications highlighting the advancements in prostate cancer treatment.

https://www.cell.com/molecular-therapy-family/nucleic-acids/fulltext/S2162-2531(24)00080-5   https://www.mdpi.com/1422-0067/25/4/2123

Our response: We appreciate the reviewer’s suggestion to add miR-34a therapeutics for mPCa patients to improve the manuscript. We have carefully reanalyzed the suggested publications and revised the manuscript to include the significance of miR-34a regulation in PCa metastasis and its relevant delivery method development. Please refer to metastasis section under ‘CRISPR technology for mPCa therapeutics’ for the updated content.

  1. A discussion on the ethical implications of using CRISPR technology in human subjects, particularly concerning potential off-target effects and long-term consequences of genetic modifications would provide a more balanced view of the technology's application.

Our response: We understand the concerns surrounding the use of CRISPR technology in humans, particularly regarding potential negative effects from off-target modifications and the long-term consequences of genetic changes. After thorough consideration, we have explained how newer CRISPR technologies can address these ethical issues. The manuscript has been updated to clarify this in the section of ‘Conclusions and future directions’.

  1. The article appears to end abruptly in the middle of a section about DNA repair inhibition, suggesting that some content may be missing or incomplete.

Our response: Regarding the concern about incompleteness of DNA repair inhibition section, we respectfully disagree with the reviewer’s suggestion. Given the limited number of reports on the effects of DNA repair gene-targeted therapy in metastatic prostate cancer (mPCa), we have concluded the section with the current information available. We believe that the manuscript includes the most relevant and up-to-date studies, and in the revision process, we have reorganized the writing to ensure a smoother transition. We hope this clarification is clear and understandable to the reviewer.

  1. There are some typographical errors and instances of awkward phrasing in the manuscript. The authors should carefully proofread to correct these issues and ensure that all abbreviations are properly defined upon first use. For example, the title of Section 5.1 could also be revised for clarity. 

Our response: We agree with the reviewer’s assessment regarding clarity of subtitles. To improve this, we have revised the subtitles to make them clearer for readers.

Reviewer 2 Report

Comments and Suggestions for Authors

I have thoroughly investigated the present manuscript entitled “CRISPR/Cas9 technology providing a therapeutic landscape of metastatic prostate cancer” by Jieun Park . The author have describe the CRISPR)/Cas9 genetic engineering tool can accelerate the effects of TI by enhancing targeted gene therapy or immunotherapy. This review describes the genetic backgrounds of metastatic PCa and how CRISPR/Cas9 technology can contribute to the field of PCa treatment development. Before acceptance the manuscript needs substantial changes to warrant the publication in this reputed journal. I suggest the following changes in order to improve the scientific quality and clarity of the present work.

1.     What is the primary mechanism by which prostate cancer (PCa) metastasizes to bone tissue, and how does this contribute to the progression of metastatic castration-resistant prostate cancer (mCRPC)? Explain with reason

2.     Author mentions the TMPRSS2-ERG gene fusion so how does the TMPRSS2-ERG gene fusion influence androgen receptor (AR) signaling in prostate cancer, and what is its significance in the development of metastatic prostate cancer (mPCa)?

3.     What role do genetic mutations in DDR pathway-related genes, such as BRCA2 and ATM, play in the aggressiveness of prostate cancer and patient response to therapies like PARP inhibitors?

4.     In section 2 describe the impact of AR amplification on the progression of castration-resistant prostate cancer (CRPC) and its correlation with resistance to androgen deprivation therapy (ADT).

5.     What is the mechanism by which AR-independent enzalutamide-resistant mechanisms bypass AR signaling in prostate cancer cells? Explain with reason

6.     In what ways do PARP inhibitors contribute to the death of HR-deficient prostate cancer cells? Explain

7.     What are the therapeutic strategies that utilize CRISPR technology to enhance the response to prostate cancer treatments or overcome drug resistance?

8.     What role do CRISPR/Cas9 screenings play in identifying novel targets for overcoming drug resistance in metastatic prostate cancer (mPCa), and how do these targets enhance treatment efficacy? Explain the reason behind this

9.     How does the loss of CHEK2 influence the resistance to PARP inhibitors in prostate cancer cells, and what implications does this have for future therapeutic strategies?

10.  What novel insights were gained from CRISPR/Cas9 screenings that identified CDK12 as a potential therapeutic target for overcoming AR signaling inhibition in prostate cancer? Explain

11.  In section 6 the author mention about the CRISPR tech. so how does CRISPR technology contribute to the discovery of metastasis drivers, such as CITED2 and KMT2C, and what is their significance in prostate cancer progression and metastasis?

12.  What are the potential benefits and limitations of using CRISPR delivery systems, such as AAV and PEI-derived graphene quantum dots (PEI-GQD), for targeted cancer therapies in metastatic prostate cancer? Explain and also include in the manuscript.

13.  The conclusion part should be refine and rewrite.

14.  There are very small typo and grammatical mistakes in whole manuscript . Please correct it.

15.  The overall English and 'spell-checked' and 'grammar-checked' needs to improve.

Based on the above comments, I would suggest minor revision for this present manuscript.

Comments on the Quality of English Language

The overall English and 'spell-checked' and 'grammar-checked' needs to improve.

Author Response

  1. What is the primary mechanism by which prostate cancer (PCa) metastasizes to bone tissue, and how does this contribute to the progression of metastatic castration-resistant prostate cancer (mCRPC)? Explain with reason.

Thank you for your insightful comment. We have revised the manuscript for giving more clarity to readers about the mechanism of PCa metastasis to other organs and its contribution to hormone resistance PCa development in the section of ‘Biology of metastatic PCa’ [1]

  1. Author mentions the TMPRSS2-ERG gene fusion so how does the TMPRSS2-ERG gene fusion influence androgen receptor (AR) signaling in prostate cancer, and what is its significance in the development of metastatic prostate cancer (mPCa)?

For information on the relationship between TMPRSS2-ERG gene fusion and AR signaling, please refer to the 'Genetics of metastatic PCa' section. We have revised the sentences to enhance clarity for the readers [2].

  1. What role do genetic mutations in DDR pathway-related genes, such as BRCA2 and ATM, play in the aggressiveness of prostate cancer and patient response to therapies like PARP inhibitors?

We appreciate your valuable suggestions about the impact of DDR gene mutations to the conventional therapies. Mutations in DNA damage repair (DDR) genes like BRCA2 and ATM increase genomic instability, making prostate cancer more aggressive and often leading to poorer responses to standard therapies. These mutations also enhance the efficacy of PARP inhibitors, which selectively target tumor cells with defective homologous recombination (HR) repair, providing a therapeutic strategy for patients with DDR mutations. We included more reports in the section of ‘Genetics of mPCa’ showing the effect of germline BRCA1/2 and CHEK2 mutations in ADT [3].

  1. In section 2 describe the impact of AR amplification on the progression of castration-resistant prostate cancer (CRPC) and its correlation with resistance to androgen deprivation therapy (ADT).

AR amplification in castration-resistant prostate cancer (CRPC) allows cancer cells to sustain AR signaling even at low androgen levels, often observed under androgen deprivation therapy (ADT). This amplification is a key driver of resistance to ADT, as it enables cancer cells to maintain proliferative signaling despite reduced androgen availability. Please refer to the section of ‘Genetics of metastatic PCa’ [4].

  1. What is the mechanism by which AR-independent enzalutamide-resistant mechanisms bypass AR signalling in prostate cancer cells? Explain with reason

Some prostate cancer cells develop resistance to enzalutamide by bypassing AR signaling altogether. This can occur through alternative growth pathways, such as upregulation of glucocorticoid receptors (GR) or activation of kinase signaling pathways. These mechanisms allow cancer cells to proliferate independently of AR signaling, making AR-targeted therapies less effective. Please refer to the section of ‘Current standard treatments’ where we elaborated about the AR -independent enzalutamide-resistant mechanisms [5].

  1. In what ways do PARP inhibitors contribute to the death of HR-deficient prostate cancer cells? Explain

PARP inhibitors are effective in killing HR-deficient prostate cancer cells by exploiting synthetic lethality. In the absence of functional HR repair (as in BRCA-mutated cells), PARP inhibition prevents single-strand break repair, leading to accumulation of double-strand breaks and ultimately cell death. Please refer to the section of ‘Inhibition of DNA repair as a targeted gene therapy for mPCa’ [6].

  1. What are the therapeutic strategies that utilize CRISPR technology to enhance the response to prostate cancer treatments or overcome drug resistance?

CRISPR-based strategies can be used to knock out genes involved in drug resistance or to enhance sensitivity to therapies. For instance, targeting resistance-associated genes may improve the effectiveness of PARP inhibitors in HR-deficient tumors or overcome resistance to AR-targeted therapies. We thoroughly analyzed the literatures utilized CRISPR technology for prostate cancer treatments and drug resistance mechanisms. Please refer to the Table 2 and the section of ‘CRISPR technology for mPCa therapeutics [7]’. 

  1. What role do CRISPR/Cas9 screenings play in identifying novel targets for overcoming drug resistance in metastatic prostate cancer (mPCa), and how do these targets enhance treatment efficacy? Explain the reason behind this

CRISPR screens help identify novel genes that contribute to drug resistance, such as those involved in DNA repair or alternative signaling pathways. Targeting these genes can improve treatment outcomes by preventing cancer cells from adapting to or surviving drug treatments. Please refer to the Table 2 and the section of ‘CRISPR technology for mPCa therapeutics’ [8]. 

  1. How does the loss of CHEK2 influence the resistance to PARP inhibitors in prostate cancer cells, and what implications does this have for future therapeutic strategies?

Loss of CHEK2, a kinase involved in the DDR pathway, may confer resistance to PARP inhibitors in prostate cancer by altering DNA repair processes. Understanding CHEK2's role in PARP resistance could guide the development of combination therapies to counteract this resistance. Please refer to the sub-section of the ‘Drug resistance mechanism under CRISPR technology for mPCa therapeutics’ [7].

  1. What novel insights were gained from CRISPR/Cas9 screenings that identified CDK12 as a potential therapeutic target for overcoming AR signaling inhibition in prostate cancer? Explain

CDK12, identified through CRISPR screens, has been shown to play a role in AR signaling inhibition resistance. Targeting CDK12 could provide a therapeutic approach to overcoming resistance in CRPC, as it is associated with maintaining genome stability and regulating transcription. Please refer to the subsection ‘Drug resistance mechanism under CRISPR technology for mPCa therapeutics’ [9].

  1. In section 6 the author mention about the CRISPR tech. so how does CRISPR technology contribute to the discovery of metastasis drivers, such as CITED2 and KMT2C, and what is their significance in prostate cancer progression and metastasis?

CRISPR screens have identified genes like CITED2 and KMT2C as potential metastasis drivers. These genes are involved in regulating pathways critical for cell migration and invasion, and their dysregulation contributes to prostate cancer progression and metastasis. Please refer to the subsection ‘Metastasis drivers under CRISPR technology for mPCa therapeutics’ [10,11].

  1. What are the potential benefits and limitations of using CRISPR delivery systems, such as AAV and PEI-derived graphene quantum dots (PEI-GQD), for targeted cancer therapies in metastatic prostate cancer? Explain and also include in the manuscript.

You have raised an important question in CRISPR field regarding how to effectively deliver the CRIPSR system to humans. AAV and PEI-GQD-based CRISPR delivery systems hold promise for targeted therapy in metastatic prostate cancer. AAV provides high specificity but is limited by delivery efficiency and immune response [12]. PEI-GQD is a newer approach that can facilitate intracellular delivery but requires further development to ensure safety and targeting accuracy in clinical applications. We updated the content regarding CRIPSR delivery system in the section of ‘CRISPR technology for mPCa therapeutics’ [13].

  1. The conclusion part should be refine and rewrite.

Thank you for your suggestion. We refined and added some contents to the conclusion part.

  1. There are very small typo and grammatical mistakes in whole manuscript . Please correct it.

                  Agreed, we carefully proofread the manuscript and tried to correct all the typo and grammatical errors.

  1. The overall English and 'spell-checked' and 'grammar-checked' needs to improve.

                  Thank you for your suggestions. We received professional English editing service for improving overall English proficiency.

Based on the above comments, I would suggest minor revision for this present manuscript.

  1. Desai, M.M.; Cacciamani, G.E.; Gill, K.; Zhang, J.; Liu, L.; Abreu, A.; Gill, I.S. Trends in Incidence of Metastatic Prostate Cancer in the US. JAMA Netw Open 2022, 5, e222246, doi:10.1001/jamanetworkopen.2022.2246.
  2. Sandhu, S.; Moore, C.M.; Chiong, E.; Beltran, H.; Bristow, R.G.; Williams, S.G. Prostate cancer. Lancet 2021, 398, 1075-1090, doi:10.1016/S0140-6736(21)00950-8.
  3. Matveev, V.B.; Lyubchenko, L.; Kirichek, A. Impact of germline DNA-repair gene BRCA2 and CHEK2 mutations on time to castration resistance in patients with metastatic hormone-naive prostate cancer: A single center analysis. J Clin Oncol 2019, 37, doi:DOI 10.1200/JCO.2019.37.15_suppl.5056.
  4. Li, Y.; Yang, R.; Henzler, C.M.; Ho, Y.; Passow, C.; Auch, B.; Carreira, S.; Nava Rodrigues, D.; Bertan, C.; Hwang, T.H.; et al. Diverse AR Gene Rearrangements Mediate Resistance to Androgen Receptor Inhibitors in Metastatic Prostate Cancer. Clin Cancer Res 2020, 26, 1965-1976, doi:10.1158/1078-0432.CCR-19-3023.
  5. Alumkal, J.J.; Sun, D.; Lu, E.; Beer, T.M.; Thomas, G.V.; Latour, E.; Aggarwal, R.; Cetnar, J.; Ryan, C.J.; Tabatabaei, S.; et al. Transcriptional profiling identifies an androgen receptor activity-low, stemness program associated with enzalutamide resistance. Proc Natl Acad Sci U S A 2020, 117, 12315-12323, doi:10.1073/pnas.1922207117.
  6. Ray Chaudhuri, A.; Nussenzweig, A. The multifaceted roles of PARP1 in DNA repair and chromatin remodelling. Nat Rev Mol Cell Biol 2017, 18, 610-621, doi:10.1038/nrm.2017.53.
  7. Tsujino, T.; Takai, T.; Hinohara, K.; Gui, F.; Tsutsumi, T.; Bai, X.; Miao, C.; Feng, C.; Gui, B.; Sztupinszki, Z.; et al. CRISPR screens reveal genetic determinants of PARP inhibitor sensitivity and resistance in prostate cancer. Nat Commun 2023, 14, 252, doi:10.1038/s41467-023-35880-y.
  8. Liu, J.; Zhao, Y.; He, D.; Jones, K.M.; Tang, S.; Allison, D.B.; Zhang, Y.; Chen, J.; Zhang, Q.; Wang, X.; et al. A kinome-wide CRISPR screen identifies CK1alpha as a target to overcome enzalutamide resistance of prostate cancer. Cell Rep Med 2023, 4, 101015, doi:10.1016/j.xcrm.2023.101015.
  9. Lei, H.Q.; Wang, Z.F.; Jiang, D.G.; Liu, F.; Liu, M.L.; Lei, X.X.; Yang, Y.F.; He, B.; Yan, M.; Huang, H.; et al. CRISPR screening identifies CDK12 as a conservative vulnerability of prostate cancer. Cell Death Dis 2021, 12, doi:ARTN 740

10.1038/s41419-021-04027-6.

  1. Cai, H.; Zhang, B.; Ahrenfeldt, J.; Joseph, J.V.; Riedel, M.; Gao, Z.; Thomsen, S.K.; Christensen, D.S.; Bak, R.O.; Hager, H.; et al. CRISPR/Cas9 model of prostate cancer identifies Kmt2c deficiency as a metastatic driver by Odam/Cabs1 gene cluster expression. Nat Commun 2024, 15, 2088, doi:10.1038/s41467-024-46370-0.
  2. Arriaga, J.M.; Ronaldson-Bouchard, K.; Picech, F.; Nunes de Almeida, F.; Afari, S.; Chhouri, H.; Vunjak-Novakovic, G.; Abate-Shen, C. In vivo genome-wide CRISPR screening identifies CITED2 as a driver of prostate cancer bone metastasis. Oncogene 2024, 43, 1303-1315, doi:10.1038/s41388-024-02995-5.
  3. Hanlon, K.S.; Kleinstiver, B.P.; Garcia, S.P.; Zaborowski, M.P.; Volak, A.; Spirig, S.E.; Muller, A.; Sousa, A.A.; Tsai, S.Q.; Bengtsson, N.E.; et al. High levels of AAV vector integration into CRISPR-induced DNA breaks. Nat Commun 2019, 10, 4439, doi:10.1038/s41467-019-12449-2.
  4. Lee, B.; Gries, K.; Valimukhametova, A.R.; McKinney, R.L.; Gonzalez-Rodriguez, R.; Topkiran, U.C.; Coffer, J.; Akkaraju, G.R.; Naumov, A.V. In Vitro Prostate Cancer Treatment via CRISPR-Cas9 Gene Editing Facilitated by Polyethyleneimine-Derived Graphene Quantum Dots. Adv Funct Mater 2023, 33, doi:10.1002/adfm.202305506.
